# Diagnostic Yield of Transbronchial Cryobiopsy Guided by Radial Endobronchial Ultrasound and Fluoroscopy in the Radiologically Suspected Lung Cancer: A Single Institution Prospective Study

**DOI:** 10.3390/cancers14061563

**Published:** 2022-03-18

**Authors:** Vytautas Ankudavicius, Skaidrius Miliauskas, Lina Poskiene, Donatas Vajauskas, Marius Zemaitis

**Affiliations:** 1Department of Pulmonology, Medical Academy, Lithuanian University of Health Sciences, LT-50161 Kaunas, Lithuania; skaidrius.miliauskas@lsmuni.lt (S.M.); marius.zemaitis@lsmuni.lt (M.Z.); 2Department of Pathology, Medical Academy, Lithuanian University of Health Sciences, LT-50161 Kaunas, Lithuania; lina.poskiene@lsmuni.lt; 3Department of Radiology, Medical Academy, Lithuanian University of Health Sciences, LT-50161 Kaunas, Lithuania; donatas.vajauskas@lsmuni.lt

**Keywords:** peripheral lung lesions, lung cancer, transbronchial cryobiopsy, diagnostic values, safety

## Abstract

**Simple Summary:**

The morphological diagnosis of peripheral lung lesions is challenging due to the anatomical location of these lesions and to the biopsy techniques that are currently available. Transbronchial forceps biopsies guided by fluoroscopy alone, or additional with radial mini probe endobronchial ultrasound (RP-EBUS), were well-known standardised procedures with an unsatisfactory diagnostic yield. Transbronchial cryobiopsy is a novel technique of lung parenchyma biopsy for the diagnosis of interstitial lung diseases, which has a very high diagnostic yield (80–96%) and relatively low incidence of complications (3–26%). Due to these reasons, transbronchial cryobiopsy guided by RP-EBUS and fluoroscopy in peripheral lung lesions are under active investigation, but still, there is limited data as only a few clinical trials of transbronchial cryobiopsy are published around the world. Our study showed that transbronchial cryobiopsy is a potentially safe procedure when obtaining a larger size of samples. Furthermore, the sensitivity, specificity, positive predictive value (PPV), negative predictive value (NPV), and accuracy were 85.1%, 100%, 100%, 12.5%, and 93% for TBCB as well as 91.5%, 100%, 100%, 20.0% and 96.7% for the combined procedures, respectively.

**Abstract:**

Transbronchial cryobiopsy (TBCB) is being studied in the diagnosis of peripheral lung lesions; however, there are only a few clinical studies around the world. The aim of our study was to evaluate the diagnostic values and safety of transbronchial cryobiopsy for radiologically suspected peripheral lung cancer. The prospective clinical study was executed from September 2019 to September 2021 at a tertiary clinical centre in Lithuania. A total of 48 patients out of 102 underwent combined procedures of transbronchial forceps biopsy (TBFB) and TBCB. Diagnostic values and safety outcomes of TBFB and TBCB were analysed. The sensitivity, specificity, positive predictive value (PPV), negative predictive value (NPV) and accuracy were 72.9%, 100%, 100%, 7.7%, and 88.0% for TBFB, 85.1%, 100%, 100%, 12.5%, and 93% for TBCB, as well as 91.5%, 100%, 100%, 20.0% and 96.7% for the combined procedures, respectively, with a significantly higher accuracy for cryobiopsies compared to forceps biopsies (*p* < 0.05). The diagnostic values for transbronchial cryobiopsies were similar, irrespective of the radial mini probe endobronchial ultrasound (RP-EBUS) position, lesion size or bronchus sign, however, the sensitivity of the combined procedures in cases with RP-EBUS adjacent to the target was significantly higher compared to TBFB (86.2% vs. 64.3%, *p* = 0.016). Samples of cryobiopsies were significantly larger than forceps biopsies (34.62 mm^2^ vs. 4.4 mm^2^, *p* = 0.001). The cumulative diagnostic yield of transbronchial cryobiopsy was 80.0% after the second biopsy and reached a plateau of 84.1% after four biopsies. No severe bleeding, pneumothorax, respiratory failure or death was registered in our study. TBCB is a potentially safe procedure, which increases diagnostic values in diagnosing peripheral lung lesions compared to TBFB.

## 1. Introduction

Peripheral lung lesions are common radiological examination findings, which histological determination is considered challenging for a pulmonologist. The spectrum of these lesions varies from a benign tumour, interstitial lung disease, to lung cancer. Early diagnosis and histological confirmation of radiological suspected lung cancer are one of the most important targets of the health care system [1,2]. Adenocarcinoma is the most common histological type and currently accounts for 50% of non-small cell lung cancers (NSCLC). The shift from squamous cell cancers to adenocarcinomas, and as result, from the central to peripheral location of tumours, was caused by changing the behaviour of smoking, the invention of more modern cigarettes, and the use of filter ventilation [3,4,5]. The transbronchial lung tumour biopsy is the diagnostic standard for histological confirmation and determination of EGFR (epidermal growth factor receptor), PD-L1 (programmed death-ligand 1), and others in the era of target therapy and immunotherapy. The data from a meta-analysis showed that the overall diagnostic yield of transbronchial forceps biopsy (TBFB) under fluoroscopic control was 57%, which dropped to 34% in tumours less than 2 cm [6,7,8]. The additional use of the radial mini probe endobronchial ultrasound (RP-EBUS) increased the diagnostic yield up to 73% [9]; however, the results are not so promising in an eccentric position of the mini probe, where the diagnostic yield was only 40% [10]. Furthermore, electromagnetic navigation bronchoscopy (ENB) in combination with RP-EBUS demonstrated a very high diagnostic yield (up to 88%) in the diagnosis of peripheral lung cancer, but the availability of the ENB is limited due to the high price of equipment and single-use instruments [11]. An alternative procedure for these peripheral lesions could be a transthoracic biopsy with a very high diagnostic yield (about 90%) according to meta-analysis, but with a high incidence of complications, mainly pneumothoraxes (23–27%) [8,12,13]. Thus, there is an unmet need for effective, less invasive, and safe procedures in the diagnosis of peripheral lung lesions. Transbronchial cryobiopsy is a novel technique for lung parenchyma biopsy for diagnosis of interstitial lung diseases [14], which has a very high diagnostic yield (80–96%) and relatively low incidence of complications (3–26%) [15,16]. TBCB, guided by RP-EBUS and fluoroscopy in peripheral lung lesions, is under active investigation, but still, there is limited data, as only several clinical trials of TBCB were published around the world [17,18].

The aim of this study is to evaluate the diagnostic accuracy, sensitivity, specificity, positive and negative predictive value, safety and complications of TBCB guided by RP-EBUS and fluoroscopy in a patient population with radiologically suspected peripheral lung cancer.

## 2. Materials and Methods

The prospective clinical study was executed at the Hospital of the Lithuanian University of Health Sciences (LUHS) Kauno Klinikos, Department of Pulmonology, in the period from September 2019 to September 2021. The study was approved by Regional Biomedical Research Ethics Committee and registered in the United States National Institutes of Health trial registry ClinicalTrials.gov with identifier NCT05164445.

### 2.1. Subjects

A total of 102 patients were referred to our tertiary university hospital with peripheral lung lesion, suspected of lung cancer, detected on the chest computed tomography (CT), to whom RP-EBUS scan examinations were performed. The TBFB was performed on all of these patients (*n* = 102) by a single pulmonologist experienced in a bronchoscopy. In addition, 48 of these patients, with no contraindications, underwent a TBCB. The contraindications for cryobiopsy were defined as follows: severe hypoxemia (pO_2_ (partial pressure of oxygen) < 60 mmHg) or hypercapnia (pCO_2_ (partial pressure of carbon dioxide) > 50 mmHg), bleeding disorders, DLCOc (the diffusing capacity for carbon monoxide) < 35% of the predictive value, FEV_1_ (a forced expiratory volume in the first second) < 800 mL or FEV_1_ < 30%, failure of the RP-EBUS to detect the peripheral lung lesion after 30 min of scanning, large vessels (diameter of more than 3 mm) near the tumour on the CT scan, technically difficult to introduce cryoprobe and/or endobronchial blocker to current bronchi segment or subsegment, excessive bleeding after transbronchial forceps biopsy, which was needed for extra intervention to stop bleeding.

### 2.2. Bronchoscopy and Biopsy Procedures

Procedures were performed under general anaesthesia with midazolam, fentanyl, and myorelaxant, and intubation with an 8.5 size rigid bronchoscope (KARL STORZ Endoscope, Tuttlingen, Germany), and high-frequency jet ventilation (TwinStream™ Carl Reiner, Stuttgart, Germany) for all patients. The TBFB was performed by flexible fibrobronchoscope (BF-H190, Olympus Evis Exera III, Olympus, Tokyo, Japan) under RP-EBUS (UMS20-17S or UM-S20-20R, Olympus, Tokyo, Japan), without a guide sheath and fluoroscopy (Philips BV Pulsera Fluoroscopy C-arm, Tokyo, Japan) control with single-use standard oval biopsy forceps suitable for 2.0 mm scope channels (Olympus, Japan). The flexible fibrobronchoscope and endobronchial blocker were introduced together through the rigid bronchoscope for the TBCB. After the localisation of the lesion with RP-EBUS and fluoroscopy, the cryoprobe (ERBE CRYO2 system, Tuebingen, Germany), connected to the cryo unit (ERBE CRYO2 system, Tuebingen, Germany), was inserted through the fibrobronchoscope working channel to the biopsy site and frozen for 4–11 s to take the TBCB. Afterward, a 5–7 Fr endobronchial blocker (Olympus, Japan) balloon was inflated to stop bleeding. At the beginning of a trial, 1.7 mm and 1.9 mm cryoprobes were used, but due to the difficulty of inserting the larger size of a cryoprobe in some (especially upper lobe) bronchi segments, with the possibility of more intensive bleeding and an introduction of smaller cryoprobes in the region, 1.1 mm diameter cryoprobes were used for the TBCB of peripheral lung lesions, as well. Freezing time depended on the size of cryoprobes, e.g., if 1.1 mm cryoprobes were used, it took a longer time than if 1.7–1.9 mm cryoprobes were used, with the aim to achieve a greater size of the specimen. From 1 to 6 samples for TBCB and from 10 to 12 samples for TBFB were taken. Samples were put in two different tubes with 10% formalin solution (first tube for TBFB, second for TBCB), then it was sent to the pathologist for histological examination. The diagnostic cases were determined as follows: biopsy considered as diagnostic if the lung cancer diagnosis or metastasis was made by histology evaluation or specific non-malignant pathologic features were observed. Normal lung tissue, a fragment of the bronchi wall, and necrosis were referred to as nondiagnostic. Complications were defined as major (moderate/severe bleeding, pneumothorax, life-threatening haemorrhagic shock, and prolonged intra-procedural hypoxemia), and minor (mild bleeding). Bleeding was defined as mild—blood suctioning required for less than one minute; moderate—suctioning required for more than one minute, repeat wedging of the bronchoscope or the application of cold saline, vasoactive substances or thrombogenic agents; severe—selective intubation needed for less than 20 min [19,20].

### 2.3. Statistical Analysis

Statistical analysis was performed using the Statistical Package for the Social Sciences, version 20.9 (IBM SPSS, Chicago, USA). The Kolmogorov–Smirnov test was used to check data normality. Normally distributed data were presented as mean ± standard deviation (SD). If significant differences were detected, the differences between two independent groups were determined by Student’s *t*-test. Categorical data were analysed using the chi-square (χ^2^) test. Sensitivity was defined as: a probability of getting a positive test result in a subject with the disease; specificity—the probability of getting a negative test result in a subject without the disease; positive predictive value (PPV)—the probability of having the disease of interest in a subject with a positive test result; negative predictive value (NPV) —the probability of not having a disease in a subject with a negative test result; diagnostic yield—the ratio of the total number of patients in whom the definite morphological diagnosis was confirmed to the total number of patients undergoing the procedure. Diagnostic accuracy was measured by the area under the receiver operating characteristic (ROC) curve. Values of *p* < 0.05 were considered to indicate statistical significance.

## 3. Results

Combined TBFB and TBCB procedures were performed in 48 patients with radiologically suspected peripheral lung cancer. The mean age of these patients was 69 years, a larger part of the participants were males and most of them were smokers. Most peripheral lung lesions were located in the upper lobes, and more than half of the lesions were smaller than 3.0 cm in diameter. Table 1 represents the main characteristics of the patients.

In more than half of the cases, the RP-EBUS was located adjacent to the target. The majority of the TBCB procedures were performed with a 1.1 mm cryoprobe with a longer freezing time. The surface area of the TBCB was greater than that of the TBFB (*p* = 0.001). The assessment of transbronchial biopsies is presented in Table 2.

The definitive morphological diagnosis was confirmed in 43 cases of 48 patients who underwent combined transbronchial procedures. The most common histological types of lung cancer were adenocarcinoma (35.4%) and squamous cells carcinoma (29.2%) (Table 3). In five cases of nondiagnostic transbronchial biopsy, lung cancer was finally diagnosed after additional procedures: one case of adenocarcinoma, one case of poorly differentiated lung carcinoma, two cases of squamous cells carcinoma were confirmed by transthoracic biopsy, and one case of squamous cells carcinoma by thoracic surgical biopsy.

The data of sensitivity, specificity, PPV, NPV and accuracy of the transbronchial procedures for all patients and lung cancer patients specifically, are presented in Table 4 and Figure 1. As all transbronchial biopsy results were assumed to be a true positive, specificity and PPV were 100%. In all cases, the sensitivity, NPV and accuracy of the TBCB were higher compared to the TBFB, and combined transbronchial procedures yielded the highest results. However, due to the small sample size, only some of the results reached a significant difference.

When analysing all patients, the accuracy of the TBCB as well as of combined transbronchial procedures were higher compared to the TBFB (area under the ROC curve was 0.935 vs. 0.880, *p* = 0.027 and 0.880 vs. 0.967, *p* = 0.027, respectively). The combined transbronchial procedures have shown the tendency for higher sensitivity compared to the TBFB, as well (91.5% vs. 72.9%, *p* = 0.07). However, there was no statistically significant difference between the accuracy of the TBCB alone compared with combined transbronchial procedures (area under the ROC curve was 0.935 vs. 0.967, *p* = 0.076).

Moreover, for lung cancer patients, the accuracy of the TBCB and of combined transbronchial procedures were higher compared to the TBFB alone, as well (area under the ROC curve was 0.923 vs. 0.859, *p* = 0.001 and 0.936 vs. 0.859, *p* = 0.001; 87.2% vs. 71.8%, *p* = 0.004, respectively). The sensitivity of combined transbronchial biopsies was significantly higher compared to the TBFB (87.2% vs. 71.8%, *p* = 0.004).

The diagnostic values of the transbronchial procedures were analysed, considering the position of the RP-EBUS to the peripheral lesion, the size of the lesion as well as positive and negative bronchus signs on the CT scan (Table 5 and Table 6) for all patients and patients with lung cancer.

In all cases, the sensitivity, NPV and accuracy of the TBCB were higher compared to the TBFB, and combined transbronchial procedures yielded the highest results irrespective of the position of RP-EBUS to the target, but only a few of them reached the significant difference. The sensitivity of the combined transbronchial biopsies in cases with the RP-EBUS adjacent to the target was significantly higher compared to the TBFB in all patients and lung cancer patients (86.2% vs. 64.3%, *p* = 0.016 and 87.0% vs. 60.9%, *p* = 0.031, respectively). In all cases, the sensitivity, NPV and accuracy were higher when the RP-EBUS was located within the target compared to the RP-EBUS located adjacent to the target; however, the results did not reach statistical significance.

Furthermore, the sensitivity, NPV and accuracy did not differ according to lesion size in each group of transbronchial procedures (TBFB, TBCB, or combined). In all patients, the accuracy of the TBCB and the sensitivity of combined transbronchial procedures were higher for lesions larger than 2 cm in diameter compared to the TBFB (Table 5). In the same way, for lung cancer patients, the sensitivity and accuracy of the TBCB and combined transbronchial procedures were higher for larger lesions compared to the TBFB (Table 6).

When analysing all patients, 14 cases had negative bronchus signs, and within these cases, the accuracy of the TBFB was 75.0%, while the accuracy of combined procedures for the final diagnosis was statistically significant and increased to 91.7% (*p* < 0.05) (Table 5). There was no statistical difference in negative bronchus signs on the CT scan group for lung cancer patients. While 34 cases had positive bronchus signs, and within these cases, the sensitivity and accuracy of the TBFB were 76.9% and 80.1%, the sensitivity and accuracy of combined transbronchial procedures significantly increased up to 96.2% and 89.7%, respectively (*p* < 0.05) (Table 6).

The cumulative diagnostic yield of transbronchial cryobiopsy was 50.0% after the first sample, 80.0% after the second and third samples, and reached a plateau of 84.1% after four biopsies. Any further biopsies did not increase the cumulative diagnostic yield.

To plot the learning curve of transbronchial cryobiopsy in relation to the number of procedures performed, we fitted a diagnostic yield considering every eight cases. In these cases, the diagnostic yield plateaued after the first 30 procedures (Figure 2).

Only six complications in all the procedures were registered: moderate bleeding was registered for two (4.16%) patients after TBCB, and mild bleeding was registered for four (8.33%) patients after TBFB, as well. No severe bleeding, pneumothorax, respiratory failure or death was registered in our study.

## 4. Discussion

The morphological diagnosis of peripheral lung lesions is challenging due to the anatomical location of these lesions and to the biopsy techniques that are currently available. TBFB, guided by fluoroscopy, was a well-known standardised historical procedure for peripheral lung lesions, with an unsatisfactory diagnostic yield for many clinicians. Recent studies showed increased diagnostic values for a TBFB guided by RP-EBUS compared to the conventional TBFB [9,21], with diagnostic yields ranging from 49.4% to 92.3% [22], a sensitivity from 49% to 88% [23,24] and accuracy from 77% to 87% [25]. The sensitivity of 72.9% and accuracy of 88% for the TBFB guided by fluoroscopy and RP-EBUS in our trial was similar to previous studies, with no statistical differences regarding lesion size, bronchus sign or position of the RP-EBUS. Our study further supports the benefit of the additional use of the RP-EBUS in the navigation of biopsy tools to peripheral lung lesions. However, the diagnostic values are far enough from desirable, especially in conditions with RP-adjacent to the target and in cases of negative bronchus signs. Thus, there is a need for novel diagnostic tools, which will have higher diagnostic accuracy and value. The first prospective study of Schuhmann M. et al. showed that the use of cryoprobes for obtaining biopsies from peripheral lung lesions is feasible and, in comparison with a forceps biopsy, significantly larger samples can be obtained without affecting safety [26].

In our prospective study, the accuracy of the TBCB (ROC area is 0.935) and combined transbronchial procedures (ROC area is 0.967) were statistically higher compared to the TBFB alone (ROC area is 0.880) for all patients with peripheral lesions (*p* = 0.027 and *p* = 0.039, respectively), with no statistically significant difference between the two cryobiopsy (TBCB and combined) procedures (*p* = 0.076). Similarly, Torky et al. demonstrated a significantly higher accuracy of combined transbronchial procedures compared to the TBFB alone (ROC area of 0.91 vs. 0.84 with *p* = 0.008), whereas this statistical difference was not significant between the TBCB and the TBFB, or the TBCB and combined accuracy, respectively [27].

Analysing only lung cancer patients, we have demonstrated the same tendency of significantly higher sensitivity and accuracy for cryobiopsies compared to the TBFB. However, there are only a few studies which analysed the diagnostic values of transbronchial biopsies specifically in lung cancer patients. Nasu et al. looked for diagnostic yield, but not for sensitivity, specificity, PPV, NPV and accuracy of transbronchial procedures in lung cancer patients [28]. Due to this, it is difficult to compare our results with other authors in the lung cancer patient population. Interestingly, Nasu, with colleagues, did not find statistically significant differences between groups, although they reported a high diagnostic yield of TBFB (86.8%), TBCB (81.1%), and combined procedures (94.3%) [29]. However, the authors discussed that the inclusion of cases when cryobiopsy was performed without a guide sheet could be the explanation of such results [29]. Corresponding, in our study, all transbronchial biopsies were performed without a guide sheet.

All this together affirms that the TBCB has excellent diagnostic value, especially in combined procedures for diagnosing peripheral lung lesions compared to the TBFB alone. Still, whether combined transbronchial biopsies—which prolonged the procedure time—yielded additional benefit compared to the TBCB alone, is open, and further larger prospective studies need to be performed, as well as whether additional interventions could cause a higher risk of complications.

The explanation of better results of cryobiopsy procedures in our study could be due to the methodology in obtaining larger samples with cryobiopsies compared to forceps biopsies in comparison to some other studies. The mean sample size of cryobiopsy in our study was 34.62 mm^2^ compared to 4.4 mm^2^ for the TBFB. Studies by Nasu et al., Schuhmann et al., and Torky et al. demonstrated a significantly larger biopsy surface area with the TBCB compared to the TBFB, 14.10 mm^2^ vs. 2.62 mm^2^, 11.17 mm^2^ vs. 4.69 mm^2^, 38.6 mm^2^ vs. 12.70 mm^2^, respectively [26,27,28]. A longer freezing time and a larger number of samples could be the reasons for the larger biopsy surface area in our study. Additionally, in 58.33 % of cases with four to six cryobiopsies taken, more than half of cryobiopsies had a freezing time of 7 to 11 s. Thus, the characteristics and techniques of cryobiopsy, especially sample surface area, are crucial for the benefit of cryobiopsy compared to forceps biopsy. It should be noted that mean tumour sizes on CT scans were similar in the analysed studies [26,27,28,29]. The application of cryoprobes of different diameters could also affect the results. In most procedures, we used a 1.1 mm diameter cryoprobe and a longer freezing time, while other authors [26,27,28] used a 1.9 mm diameter cryoprobe and a shorter freezing time. The results of our study suggest that the application of a cryoprobe 1.1 mm in diameter is a feasible tool for the biopsy of peripheral lung lesions with a high diagnostic yield.

It is well-known that tumour size, the position of the RP-EBUS, and bronchus sigh are predictive factors for diagnostic values of TBFB [30,31,32,33,34,35]. In our study, the diagnostic values of transbronchial cryobiopsies in all patients and lung cancer patients were similar regardless of the position of the RP-EBUS to the lesion. Although the benefit of cryobiopsies compared to TBFBs, in cases where the RP-EBUS was adjacent to the peripheral lesion, was confirmed. However, this was not the same within the lesion. Several previous publications also reported additional benefits of using cryobiopsy compared to the TBFB, especially when the RP-EBUS was adjacent to the target [26,32,33]. These studies showed that the diagnostic yield was 83.7–92.5% for cryobiopsies, 80.0–84.0% for the TBFBs when the RP-EBUS was positioned within the lesion, 66.7–84.3% for cryobiopsy, and 48.0–62.0% for the TBFBs when the RP-EBUS was adjacent to the target, respectively [6,25,28,32,34]. Furthermore, Matsumoto et al. reported a statistically significant advantage of cryobiopsy versus conventional procedures when the R-EBUS was adjacent to the lesion (84.3% vs. 69.4%, *p* = 0.001) [32]. The results of our study were similar to the Matsumoto study when the RP-EBUS was positioned within the target (with a sensitivity and accuracy of up to 100%) and was numerically higher compared to other studies when the RP-EBUS was adjacent to the target (sensitivity of 85.7–87.0% and accuracy of 92.8–85.1%) for all patients and lung cancer patients, respectively. The samples from the lateral area are difficult to collect by forceps in cases when the RP-EBUS is adjacent to the target. On the other hand, when the cryoprobe is adjacent to the target, it enables to take an entirely circumferential cryobiopsy from the lesion, which includes the lateral area and could increase efficiency compared with forceps biopsy [32,34,35]. It should be noticed that the other authors had analysed only a few cases of the RP-EBUS adjacent to the lesion; therefore, any statistical conclusion could not be drawn [26,28].

The direct correlation between lesion size and diagnostic values for the TBFBs guided by fluoroscopy, but not guided by EBUS-RP, was described in the literature [29,30,31]. In our study, the diagnostic values of cryobiopsy were statistically insignificant regardless of the lesion size in all patients and lung cancer patients. Other authors also did not find the correlation between lesion size and diagnostic values of cryobiopsies. Nasu et al. noticed that the diagnostic yield of the TBCB was 80.8% for lesions ≤ 3 cm and 81.5% for lesions > 3 cm [28]. In addition, in the study of Matsumoto et al. no statistically significant difference in diagnostic yield was found between the lesions ≤ 2 cm (86.7%) and lesions > 2 cm (91.8%) [32]. Thus, transbronchial cryobiopsy could be a valuable option for the biopsy of small peripheral lung lesions, too.

In several studies, the bronchus sign on the CT scan is described as a significant factor, which correlates with RP-EBUS visualisation and accessibility of peripheral lesions. It also affects the overall diagnostic values of the TBFB and TBCB, as well [28,32,36,37]. Moreso, 75% of patients enrolled in the study conducted by Matsumoto et al. had a positive bronchus sign; these patients also had a significantly higher diagnostic yield in comparison, where the bronchus sign was negative (94.3% vs. 76.9%, respectively) [32]. Nasu, together with colleagues, analysed data, where almost 93% of participants had a positive bronchus sign on CT. The diagnostic yield of the TBCB was statistically higher in the group of positive bronchus signs (85.7%) compared with negative bronchus signs (25.0%) [28]. Our results were similar to Matsumoto et al. We enrolled 71% of patients who had positive bronchus signs. The sensitivity of the TBCB was 91.2% in patients with positive bronchus signs and 75.0% in patients with negative bronchus signs in all the patients. Additionally, the results were very similar in a subgroup analysis of the lung cancer patients. However, these results did not reach statistical differences. This could be explained by a low number of cases.

The cumulative diagnostic yield of the TBCB in our study was 80.0% after the second biopsy and reached a plateau of 84.1% after four biopsies. Schuhmann et al. showed similar results with an increased cumulative diagnostic yield up to 71% after the second transbronchial cryobiopsy [26]. It seems that in daily clinical practice, it is enough to take 2–4 transbronchial cryobiopsies in radiological suspected peripheral lung cancer.

Our study suggests that approximately 30 procedures are needed for a pulmonologist to acquire the desired level of expertise for transbronchial cryobiopsies in peripheral lung lesions. Almeida et al. analysed the learning curves in relation to the number of procedures, but in different patient populations. They showed that the learning curve plateaus at a diagnostic yield of 90% after approximately 70 procedures in patients with suspected diffuse lung disease [38].

The safety of the transbronchial biopsies is one of the cornerstone aspects, especially in the case of cryobiopsy. Kuse, together with colleagues, had analysed clinical records of 50 patients who underwent TBCB for the diagnosis of pulmonary diseases. No mortality associated with a TBCB, pneumothorax or severe bleeding requiring any interventions were observed, and only one patient had pneumonia, thus requiring antibiotic therapy after the TBCB [39]. Matsumoto et al. described 101 cases (39.3%) of mild, 100 cases (38.9%) of moderate, and three cases (1.2%) of severe bleeding after a TBCB, as well as two cases (0.8%) of pneumothorax [32]. Herth et al. noticed that 166 of 1024 participants (16%) had experienced some grade of bleeding, but the grade 1 bleeding was the most common of them (47%), and pneumothorax was accrued in 6.6 % of participants [40]. Imabayashi, with colleagues from Japan, also analysed complications after a TBCB: grade 1 bleeding occurred in 25 patients (71.4%) and in grade 2, only one patient (2.8%), and pneumothorax occurred in two patients (5.7%) [41]. Nasu et al. reported only one severe bleeding; Schuhmann reported about one moderate bleeding after a TBCB and no other bleeding or pneumothorax were noticed after procedures in both studies [26,28]. Our data showed a very low rate of complications, as well. We registered only two cases (4.16%) of moderate bleeding after a TBCB. Furthermore, no mild or severe bleeding, pneumothorax, respiratory failure or death was registered in our study at all. A possible explanation for this, maybe that, in our study, the rigid bronchoscopy and endobronchial blockers were used during all TBCB procedures, like in Kuse or Herth’s studies [39,40]. However, the results are very controversial, because some other authors did not use the endobronchial blocker or used them only in some cases and noticed a higher [32,40,41] or similar [26,28,39] incidence of bleeding compared with our trial. Although there is no question that the endobronchial blocker effectively controls bleeding after a TBCB, other factors like the diameter of the cryoprobes or freezing time could also influence the results. We used a 1.1 mm diameter cryoprobe in most of the procedures, similar to Schuhmann et al., who used a 1.2 mm diameter cryoprobe [26]; while others used 1.9–2.4 mm diameter cryoprobes [26,32,38,39]. It should be noticed that the mean freezing time in our study was 7.6 s, while the other authors described a shorter freezing time (3–5 s) [26,27,28,32,39,40,41]. Authors used a different classification of complications, as well [26,28,41,42]. For all our patients, an RP-EBUS examination was performed, similarly as in previously reported studies, which helped to detect large vessels near the lesion and avoid them during the biopsy [21,26,39,40].

## 5. Limitations

Our study had several limitations. First, it was performed at a single tertiary centre with a relatively low number of patients. Second, we could not make a definite conclusion about the appropriateness of TBLC in small peripheral lung lesions due to a low number of cases. Further, larger trials are required to establish conclusions. Third, we selectively performed cryobiopsy after forceps biopsy, which may have skewed our data towards an increased diagnostic value for cryobiopsy. In some situations, the operator failed to successfully navigate to the peripheral lung lesion, and these cases were excluded from our study. Thus, it could have potentially introduced a selection bias towards easier lesions affecting the overall diagnostic values. Bronchoscopists had a high level of experience for interventional bronchoscopy, including rigid bronchoscopy, and transbronchial biopsies.

## 6. Conclusions

Transbronchial cryobiopsy can be safely conducted with a rigid bronchoscopy and bronchial blocker and allows for the collection of larger specimens. When combined with transbronchial forceps biopsy, it may improve the diagnostic accuracy of peripheral lung nodules, including nodules with an eccentric view on radial probe ultrasound. Our findings suggest that an optimal number of cryobiopsies are 2–4 samples, and on average, 30 procedures are needed for a pulmonologist to acquire the desired level of expertise in transbronchial cryobiopsies. No severe complications were registered in our study.

## Figures and Tables

**Figure 1 cancers-14-01563-f001:**
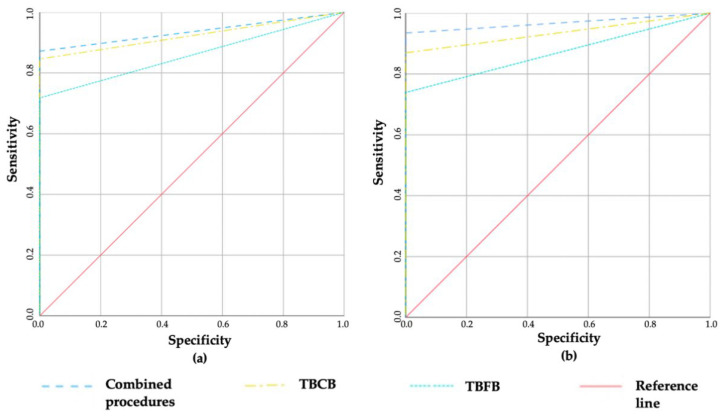
Diagnostic accuracy of procedures. (**a**) ROC curve of all patients, (**b**) ROC curve of patients with lung cancer, TBCB—transbronchial cryobiopsy, TBFB—transbronchial forceps biopsy, *p* = 0.001.

**Figure 2 cancers-14-01563-f002:**
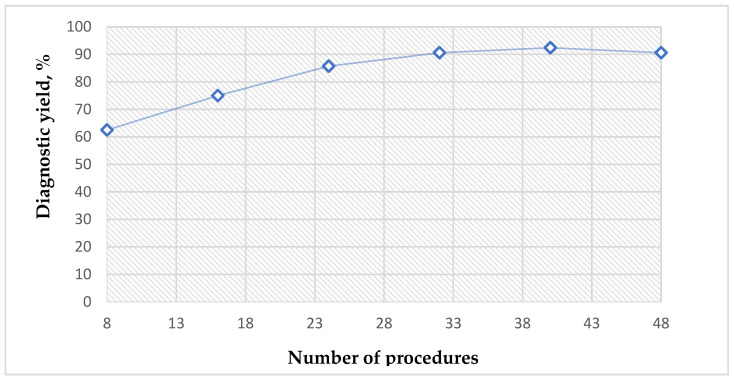
Diagnostic yield of transbronchial cryobiopsy in relation to the number of procedures performed in all patients. %—percentage.

**Table 1 cancers-14-01563-t001:** Characteristics of the patients.

Category	Value
Age, mean ± SD, years	69.35 ± 8.58
Gender, *n* (%)	
Male	32 (66.7)
Female	16 (33.3)
Smoking status, *n* (%)	
Current smoker	12 (25.0)
Ex-smoker	24 (50.0)
Non-smoker	12 (25.0)
Lesion size, mean ± SD, cm	3.52 ± 2.07
>3.0 cm	21 (43.7)
>2.0 ≤ 3 cm	16 (33.4)
≤2.0 cm	11 (22.9)
Lobar location, *n* (%)	
Upper	28 (58.3)
Middle	3 (6.3)
Lower	17 (35.4)
Bronchus sign, *n* (%)	
negative	14 (29.2)
positive	34 (70.8)

*n*—number of patients, SD—standard deviation, %—percentage, cm—centimetres.

**Table 2 cancers-14-01563-t002:** Assessment of transbronchial biopsies.

Category	Value
RP-EBUS, *n* (%)	
adjacent the target	29 (60.4)
within the target	19 (39.6)
Diameter of cryoprobe, *n* (%)	
1.1 mm	31 (64.6)
1.7 mm	3 (6.3)
1.9 mm	14 (29.2)
Number of cryobiopsy samples, *n*	
Mean ± SD	3.5 ± 0.9
Median (min; max)	4 (1; 6)
Cryobiopsy freezing time (s)	
Mean ± SD	7.6 ± 2.3
Median (min; max)	7 (4; 13)
Biopsy surface area, mean ± SD, mm^2^	
TBFB	4.47 ± 1.87
TBCB *	34.62 ± 13.65

TBCB—transbronchial cryobiopsy, TBFB—transbronchial forceps biopsy, RP-EBUS—radial mini probe endobronchial ultrasound, *n*—number of cases, SD—standard deviation, %—percentage, cm—centimetres, mm—millimetres, * *p* = 0.001 TBCB compared with TBFB.

**Table 3 cancers-14-01563-t003:** Morphological diagnosis of transbronchial biopsies for peripheral lung lesions.

	Value, *n* (%)
**Lung cancer**	
Adenocarcinoma	17 (35.4)
Squamous cells carcinoma	14 (29.2)
Adenosquamous cells carcinoma	1 (2.1)
Small cells lung carcinoma	3 (6.3)
**Other**	
Metastasis	4 (8.3)
Nonspecific inflammation	1 (2.1)
Granulomatous inflammation	2 (4.2)
Lymphoma	1 (2.1)
**Nondiagnostic biopsy**	
Normal lung tissue	4 (8.3)
Fragment of bronchi wall	1 (2.1)

*n*—number of cases, %—percentage.

**Table 4 cancers-14-01563-t004:** Sensitivity, negative predictive value (NPV), accuracy, positive predictive value (PPV), and specificity for all patients and patients with lung cancer.

	Sensitivity (%)	NPV (%)	Accuracy (%)	PPV (%)	Specificity (%)
**All patients**					
TBFB	72.9	7.7	88	100	100
TBCB	85.1	12.5	93.5 ^b^	100	100
Combined procedures	91.5 ^a^	20	96.7 ^c,d^	100	100
**Patients with lung cancer**					
TBFB	71.8	40	85.9	100	100
TBCB	84.6	53.3	92.3 ^f^	100	100
Combined procedures	87.2 ^e^	57.1	93.6 ^g^	100	100

TBCB—transbronchial cryobiopsy, TBFB—transbronchial forceps biopsy, RP-EBUS—radial mini probe endobronchial ultrasound, NPV—negative predictive values, PPV—positive predictive value, ^a^—*p* = 0.070 combined procedures compared with TBFB, ^b^—*p* = 0.027 TBCB compared with TBFB, ^c^—*p* = 0.039 combined procedures compared with TBFB, ^d^—*p* = 0.076 combined procedures compared with TBCB, ^e^—*p* = 0.004 combined procedures compared with TBFB, ^f^—*p* = 0.001 TBCB compared with TBFB, ^g^—*p* = 0.001 combined procedures compared with TBFB.

**Table 5 cancers-14-01563-t005:** Subgroup analysis of sensitivity, negative predictive value (NPV), and accuracy for all patients.

Category	TBFB			TBCB			Combined Procedures		
	(*n* = 48)			(*n* = 48)			(*n* = 48)		
	Sensitivity (%)	NPV (%)	Accuracy (%)	Sensitivity (%)	NPV (%)	Accuracy (%)	Sensitivity (%)	NPV (%)	Accuracy (%)
RP-EBUS									
Adjacent the target	64.3	9.1	82.1	85.7 ^a^	20	92.8	86.2 ^b^	25	94.6
Within the target	88.9	33.3	94.4	94.4	50	97.2	100	100	100
Lesion size									
≤3.0 cm	73.1	12.5	86.5	88.5	25	94.2	92.3 ^c^	33.3	96.2
>3.0 cm	75	16.7	87.5	85	25	92.5	95	50	97.5
≤2.0 cm	72.7	0	N/A	90.9 ^d^	0	N/A	100	0	N/A
>2.0 cm	74.3	18.2	87.1	85.7	28.6	92.9 ^e^	91.4 ^f^	40	95.7 ^g^
Bronchus sign									
Negative	58.3	28.6	75	75	40	87.5 ^h^	83.3	50	91.7 ^j^
Positive	79.4	0	N/A	91.2	0	N/A	100 ^i^	0	N/A

TBCB—transbronchial cryobiopsy, TBFB—transbronchial forceps biopsy, RP-EBUS—radial mini probe endobronchial ultrasound, NPV—negative predictive values, PPV—positive predictive value, *n*—number of patients, %—percentage, cm—centimetres, N/A—the area under the ROC curve was not displayed, ^a^—*p* = 0.07 TBCB compared with TBFB, ^b^—*p* = 0.016 combined procedures compared with TBFB, ^c^—*p* = 0.063 combined procedures compared with TBFB, ^d^—*p* = 0.28 TBCB compared with TBFB, ^e^—*p* = 0.031 TBCB compared with TBFB, ^f^—*p* = 0.021 combined procedures compared with TBFB, ^g^—*p* = 0.031 combined procedures compared with TBFB, ^h^—*p* = 0.07 TBCB compared with TBFB, ^j^—*p* = 0.07 combined procedures compared with TBFB, ^i^—*p* = 0.002 combined procedures compared with TBFB.

**Table 6 cancers-14-01563-t006:** Subgroup analysis of sensitivity, negative predictive value (NPV), and accuracy for patients with lung cancer.

Category	TBFB			TBCB			Combined Procedures		
	(*n* = 40)			(*n* = 40)			(*n* = 40)		
	Sensitivity (%)	NPV (%)	Accuracy (%)	Sensitivity (%)	NPV (%)	Accuracy (%)	Sensitivity (%)	NPV (%)	Accuracy (%)
RP-EBUS									
Adjacent the target	60.9	18.2	72.1	87	40	85.1	87.0 ^a^	25	85.8
Within the target	93.8	66.7	96.9	93.8	50	96.9	100	100	100
Lesion size									
≤3.0 cm	69.6	25	84.8	82.6	25	78.8	87	33.3	81
>3.0 cm	75	33.3	87.5	87.5 ^b^	50	93.8	93.8 ^c^	50	96.9 ^d^
≤2.0 cm	77.8	33.3	88.9	88.9 ^e^	25	69.4 ^f^	100	18.2	75
>2.0 cm	70	27.3	85	83.3	42.9	91.7 ^g^	86.7 ^h^	40	93.3 ^j^
Bronchus sign									
Negative	63.6	42.9	81.8	72.7	40	86.4	81.8	50	90.9
Positive	76.9	14.3	80.1	92.3	33.3	87.8	96.2 ^i^	17.6	89.7 ^k^

TBCB—transbronchial cryobiopsy, TBFB—transbronchial forceps biopsy, RP-EBUS—radial mini probe endobronchial ultrasound, NPV—negative predictive values, PPV—positive predictive value, *n*—number of patients, %—percentage, cm—centimetre, N/A—the area under the ROC curve was not displayed, ^a^—*p* = 0.031 combined procedures compared with TBFB, ^b^—*p* = 0.02 TBCB compared with TBFB, ^c^—*p* = 0.031 combined procedures compared with TBFB, ^d^—*p* = 0.035 combined procedures compared with TBFB, ^e^—*p* = 0.34 TBCB compared with TBFB, ^f^—*p* = 0.23 TBCB compared with TBFB, ^g^—*p* = 0.031 combined procedures compared with TBFB, ^h^—*p* = 0.002 TBCB compared with TBFB, ^j^—*p* = 0.036 combined procedures compared with TBFB, ^i^—*p* = 0.01 combined procedures compared with TBFB, ^k^—*p* = 0.003 combined procedures compared with TBFB.

## Data Availability

The data presented in this study are available on request from the corresponding author.

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
