# Peer review of "Diagnostic Yield of Transbronchial Cryobiopsy Guided by Radial Endobronchial Ultrasound and Fluoroscopy in the Radiologically Suspected Lung Cancer: A Single Institution Prospective Study"

_cancers, 2022, doi:10.3390/cancers14061563_

Round 1

Reviewer 1 Report

Dear Authors:

Thank you for submitting your manuscript for consideration for publication. Please find below my comments:

Major:

  • There is a significant concern for selection bias in the design of the study. It is unclear in the manuscript how authors have the selected the 48 patients out of the total of 102 to undergo TBCB in addition to the TBFB. Was it randomized? Was there another strategy involved?
  • During statistical analysis comparing the sensitivity and accuracy of TBFB and TBCB, what was the sample size used? Where the 48 TBCB procedures compared to the 102 total TBFB procedures? or was the comparator an internal control (i.e TBFB and TBCB were compared within the same 48 patients)?
  • It is mentioned that "All transbronchial biopsy results were assumed to be true positives ...". Why was that assumption made?
  • The largest challenge in nodule sampling and diagnostic yield is usually with the smaller nodules (<2.0 cm). Did the authors conduct a comparison in diagnostic yield between TBFB and TBCB for lesions < 2.0 cm? Can they provide a p values?
  • Can the authors clarify how they navigated to the target nodule? what navigation platform was used (ultrathin bronchoscopy, electromagnetic bronchoscopy, virtual bronchoscopy, robotic, etc)? Why was a guide sheath not used in this study?
  • There has been a number of navigation failures (unable to reach the target nodule). We recommend that the authors comments on the percentage of navigation failure and how this could lead to selection bias.
  • The complications is this study were unexpectedly low compared to what has been published in the literature. How did authors clarify bleeding as mild/moderate/severe? what definition was used?
  • The conclusion needs some editing to be done. The authors state that the TBCB helps with eccentric lesions. Has there been a direct comparison between TBCB in eccentric and concentric lesions? what was the p-value?

Minor:

  • We recommend enhancements to be made to the English grammar and phrases structures.
  • We recommend that a paragraph on "Limitations" be added to the manuscript. Limitations include selection bias, lack of generalizability (single bronchoscopist for all procedures), the use of rigid bronchoscopy (the vast majority of bronchoscopists globally are not trained on rigid bronchoscopy), not accounting for navigation failures, tec.

Thank you

Author Response

Dear reviewer,

Thank you for your comments.

Major:

  • There is a significant concern for selection bias in the design of the study. It is unclear in the manuscript how authors have the selected the 48 patients out of the total of 102 to undergo TBCB in addition to the TBFB. Was it randomized? Was there another strategy involved?

From line 97 to 107 we described exclusion criteria for cryobiopsy. Thus, 54 of 102                 patients had these contraindications.  48 of 102 patients had no contraindications and      were randomized in our trial. For 48 patients TBFB and TBCB were performed as             well.

  • During statistical analysis comparing the sensitivity and accuracy of TBFB and TBCB, what was the sample size used? Where the 48 TBCB procedures compared to the 102 total TBFB procedures? or was the comparator an internal control (i.e TBFB and TBCB were compared within the same 48 patients)?

In section “3. Results” (line 156-157) we described, that TBFB and TBCB procedures were performed for 48 patients. So statistical analysis of TBFB and TBCB were compared within the same 48 patients.

  • It is mentioned that "All transbronchial biopsy results were assumed to be true positives ...". Why was that assumption made?

Corrected.

As all positive transbronchial biopsy results were assumed to be true positive, specificity and PPV were 100% - the positive transbronchial biopsy results were not confirmed futher by any other method.  

  • The largest challenge in nodule sampling and diagnostic yield is usually with the smaller nodules (<2.0 cm). Did the authors conduct a comparison in diagnostic yield between TBFB and TBCB for lesions < 2.0 cm? Can they provide a p values?

Yes, we compared the diagnostic values between TBFB and TBCB for lesions < 2.0 cm, but no statistically significant results were observed (p>0.05). The data are presented in the Table 5/6 and only statistically significant results were marked.

  • Can the authors clarify how they navigated to the target nodule? what navigation platform was used (ultrathin bronchoscopy, electromagnetic bronchoscopy, virtual bronchoscopy, robotic, etc)? Why was a guide sheath not used in this study?

The procedures were performed with small size bronchoscope and navigated by fluoroscopy and radial mini probe endobronchial ultrasound (RP-EBUS). Our study was designed to perform procedures without guide sheath.

  • There has been a number of navigation failures (unable to reach the target nodule). We recommend that the authors comments on the percentage of navigation failure and how this could lead to selection bias.

Only cases with sucsesfully navigation were enrolled in our study. The number of navigation failures were not calculated.

  • The complications is this study were unexpectedly low compared to what has been published in the literature. How did authors clarify bleeding as mild/moderate/severe? what definition was used?

It could be explained by different classification of bleeding and technique of procedures. More explanations, you can find in the line 415-423.

Bleeding definition already added in the line 136-140.

  • The conclusion needs some editing to be done. The authors state that the TBCB helps with eccentric lesions. Has there been a direct comparison between TBCB in eccentric and concentric lesions? what was the p-value?

Yes, we compared diagnostic values between TBCB in eccentric and concentric position, but no statistically significant results were observed (p>0.05).

However, we mean, that statistically significant benefit for cryobiopsies compared to TBFB, in cases where the RP-EBUS were adjacent to peripheral lesion, was confirmed (p values presented in the Table 5 and Table 6).

The conclusion was edited.

Minor:

  • We recommend enhancements to be made to the English grammar and phrases structures.

Corrected.

  • We recommend that a paragraph on "Limitations" be added to the manuscript. Limitations include selection bias, lack of generalizability (single bronchoscopist for all procedures), the use of rigid bronchoscopy (the vast majority of bronchoscopists globally are not trained on rigid bronchoscopy), not accounting for navigation failures, tec.

Limitations already added to the manuscript as you recommended (line 431-439).

Sincerely,

 Authors.

Reviewer 2 Report

Thank you for your paper; it is innovative and very interesting.

In "Bronchoscopy and Biopsy Procedure" section, the complications are divided in major and minor; but in "result" section the author describe moderate bleeding, mild bleeding and severe bleeding. I don't find the definition of mild, moderate and severe. Even in the use of ILD cryobiopsy the problem is much discussed, so it is important to define well how bleeding is defined. 

A critical bias of the paper, it's the use of different cryo-probe. 17 procedure are performed with 1.7 mm or 1.9mm probe, this implies a different cooling time, different simple size. Please clarify.

The majority of the lesion are > 20mm, more simple to reach. The table 5 is incomprehensible. 

The conclusion that at least 30 TBCB are needed maybe is difficult to demonstrate. It's only a personal experience.Please clarify.

Have you performed the molecular analyzes on sample take? And if no why?

Author Response

Dear reviewer,

Thank you for your comments.

In "Bronchoscopy and Biopsy Procedure" section, the complications are divided in major and minor; but in "result" section the author describe moderate bleeding, mild bleeding and severe bleeding. I don't find the definition of mild, moderate and severe. Even in the use of ILD cryobiopsy the problem is much discussed, so it is important to define well how bleeding is defined. 

Bleeding definition already added in the line 136-140.

A critical bias of the paper, it's the use of different cryo-probe. 17 procedure are performed with 1.7 mm or 1.9mm probe, this implies a different cooling time, different simple size. Please clarify.

Yes, but only 3 of 17 procedures were performed by 1.7 mm cryoprobe and it did not change our results and conclusions. Also, the diagnostic values and other parameters are not divided by cryoprobe size, it is only used to describe the basic features of the data in the study.

The majority of the lesion are > 20mm, more simple to reach. The table 5 is incomprehensible. 

A lot of studies analyses diagnostic values for peripheral lesions, so the data on the Table 5 help us to compare and discus our results to these studies. Additionally, we looked specifically  to the lung cancer patient's group, which is not so represented in the literature.

The conclusion that at least 30 TBCB are needed maybe is difficult to demonstrate. It's only a personal experience.Please clarify.

Yes, we agree, it is based on personal experience in our prospective single institution study.

To confirm this conclusion, we fitted diagnostic yield considering every eight cases. In these cases, the diagnostic yield plateaued after the first 30 procedures. Also, the data is presented in the Figure 2. These results support our conclusion, that average 30 procedures are needed for a pulmonologist to acquire the desired level of expertise in transbronchial cryo-biopsies.

Have you performed the molecular analyzes on sample take? And if no why?

Yes, the expression of PD-L1 was performed for all patients with the lung cancer and EGFR, ALK were performed for patients with  adenocarcinoma as routine clinical practice in our hospital.  The results of the molecular analyzes will be publish in the future.

Sincerely,

Authors.

Round 2

Reviewer 1 Report

Dear Authors:

Thank you for your reply and for addressing prior comments. The manuscript reads better. Please find below my comments below:

1- The paragraph on “Limitations” looks good. I recommend that the authors add one more limitation related to the exclusion of cases where the operator failed to successfully navigate to the target nodule as this could have potentially introduced a selection bias towards easier nodules affecting the overall diagnostic yield.

2- Please include p-values in the Table5/6 for comparisons made between TBFB and TBCB for nodules ≤ 2.0 cm.

3- Please clearly state in the “Materials and Methods” section that navigation was done without a guide sheath.

4- The “Conclusions” section needs editing. It reads stronger that the actual evidence provided by this study. I recommend that you consider the following alternative: Instead of stating that “Transbronchial cryobiopsy is a potentially safe procedure …”, you may consider something like “Transbronchial cryobiopsy can be safely conducted with rigid bronchoscopy and bronchial blocker and allows for collection of larger specimens. When combined with transbronchial forceps biopsy, it may improve diagnostic accuracy of peripheral lung nodules including nodules with an eccentric view on radial probe ultrasound. Our findings suggest that …”.

Thank you

Author Response

Thank you for your comments.

1- The paragraph on “Limitations” looks good. I recommend that the authors add one more limitation related to the exclusion of cases where the operator failed to successfully navigate to the target nodule as this could have potentially introduced a selection bias towards easier nodules affecting the overall diagnostic yield.

Corrected. Line 450-451.

2- Please include p-values in the Table5/6 for comparisons made between TBFB and TBCB for nodules ≤ 2.0 cm.

Corrected. Please look at the Table 5/6

3- Please clearly state in the “Materials and Methods” section that navigation was done without a guide sheath.

Corrected. Line 124.

4- The “Conclusions” section needs editing. It reads stronger that the actual evidence provided by this study. I recommend that you consider the following alternative: Instead of stating that “Transbronchial cryobiopsy is a potentially safe procedure …”, you may consider something like “Transbronchial cryobiopsy can be safely conducted with rigid bronchoscopy and bronchial blocker and allows for collection of larger specimens. When combined with transbronchial forceps biopsy, it may improve diagnostic accuracy of peripheral lung nodules including nodules with an eccentric view on radial probe ultrasound. Our findings suggest that …”.

Conclusions were corrected.

Sincerely,

Authors.